# Fresh Compost Tea Application Does Not Change Rhizosphere Soil Bacterial Community Structure, and Has No Effects on Soybean Growth or Yield

**DOI:** 10.3390/plants10081638

**Published:** 2021-08-10

**Authors:** Rana Bali, Jonathan Pineault, Pierre-Luc Chagnon, Mohamed Hijri

**Affiliations:** 1Institut de Recherche en Biologie Végétale (IRBV), Université de Montréal, 4101 Rue Sherbrooke Est, Montréal, QC H1X 2B2, Canada; rana.bali@umontreal.ca; 2Écomestible Inc., 470 Rue Constable, McMasterVille, QC J3G 1N6, Canada; jonathan@ecomestible.com; 3African Genome Center, Mohammed VI Polytechnic University (UM6P), Lot 660, Hay Moulay Rachid, Ben Guerir 43150, Morocco

**Keywords:** conventional agriculture, sustainable agriculture, compost tea, bacteria, biodiversity, illumina MiSeq sequencing, plant growth, yield, soybean

## Abstract

Soil bacteria drive key ecosystem functions, including nutrient mobilization, soil aggregation and crop bioprotection against pathogens. Bacterial diversity is thus considered a key component of soil health. Conventional agriculture reduces bacterial diversity in many ways. Compost tea has been suggested as a bioinoculant that may restore bacterial community diversity and promote crop performance under conventional agriculture. Here, we conducted a field experiment to test this hypothesis in a soybean-maize rotation. Compost tea application had no influence on bacterial diversity or community structure. Plant growth and yield were also unresponsive to compost tea application. Combined, our results suggest that our compost tea bacteria did not thrive in the soil, and that the positive impacts of compost tea applications reported elsewhere may be caused by different microbial groups (e.g., fungi, protists and nematodes) or by abiotic effects on soil (e.g., contribution of nutrients and dissolved organic matter). Further investigations are needed to elucidate the mechanisms through which compost tea influences crop performance.

## 1. Introduction

Soil bacteria drive key ecosystem functions, including litter decomposition, nutrient mobilization, crop protection against pathogens and soil aggregation [1,2,3,4]. Bacterial species are not functionally redundant, which translates into positive correlations between ecosystem functioning and bacterial diversity [5,6,7]. As a result, bacterial diversity is now recognized as a component of soil health and a central issue in the development of sustainable agriculture practices [2,8,9].

Conventional agriculture can negatively influence bacterial diversity, community structure and biomass in many ways. Tillage has been widely reported to negatively affect bacterial diversity in croplands [10,11,12,13]. The same is true for chemical fertilizers that have been shown to reduce soil functional diversity [14], and lead to community dominance by a few taxa [15,16,17,18]. Pesticides also reduce soil microbial diversity and enzymatic activity, which may compromise soil health and plant performance in the long run [19,20,21]. Various strategies have been suggested to alleviate the negative effects of these conventional practices. Among these, plant biostimulants such as microbial-based inoculants are deemed a promising solution for improvement of plant performance and ecosystem functioning [22,23,24,25,26,27].

Numerous microbial inoculants have been developed for organic and conventional farming [28,29,30,31]. Mycorrhizal fungi and nitrogen-fixing bacteria, for example, are widely used to promote plant growth and soil fertility in harsh conditions [32]. Likewise, various rhizobacteria have been found to promote plant growth and vigor [33]. However, the positive impacts of these microbial inoculants are likely to be context-dependent [34], and unlikely to restore microbial diversity in agricultural soils. Alternative solutions that better promote microbial diversity, and thus ecosystem function [7] and resilience [35], should therefore be explored. Compost tea has been suggested as one such solution [36,37].

Compost tea is an inoculant prepared through aerobic, liquid-based incubation of compost amended with carbon sources. This promotes microbial proliferation [36,37,38]. Over a short incubation time (typically 48 h), high cell densities can be achieved, allowing the application of a diluted suspension over large surfaces. Aerated compost tea is presumed to be an environmentally safe product that could enhance crop performance, in part, through the reintroduction of diverse soil bacteria contributing to nutrient cycling [37,38].

Positive yield responses to compost tea have been reported for a variety of crops [39,40,41]. Based on such findings, many authors have concluded that compost tea treatment could be used as a plant growth-promoting technique in organic cultivation of crops. However, we still lack basic information on the mechanisms through which compost tea influences indigenous microbial communities and crop yields. Specifically, we still do not know whether bacteria inoculated through compost tea can survive, successfully establish themselves and compete against indigenous bacterial communities in the soil. Further studies are required to trace and track changes in microbial communities following compost tea application, with appropriate experimental controls, in order to verify the potential impacts of the tea on crop growth and yield. We also need to better distinguish the biotic effects of applying compost tea to soil (i.e., its contribution of beneficial biotas) from its abiotic effects (i.e., its addition of nutrients to the soil in mineral and dissolved organic forms). In this study, we conducted a field experiment to evaluate the impact of compost tea application on bacterial community structure and crop performance in a conventional soybean monoculture system. We hypothesized that compost tea application would promote bacterial diversity, and more specifically, Proteobacteria, which are commonly regarded as opportunistic copiotrophs [42], that should capitalize on simple sugars included as amendments during compost tea preparation. We anticipated that this shift in bacterial communities would, in turn, improve soybean growth and yield, given the wide range of known plant growth-promoting taxa among Proteobacteria.

## 2. Results

### 2.1. Bacterial Community Composition

From our total of 119 samples (108 soil samples and 11 compost tea samples), we retrieved 737 bacterial ASVs belonging to 13 phyla. To determine patterns of bacterial richness, we performed ASV rarefaction analysis for all samples, which showed that our sequencing depth was appropriate since all curves reached an asymptote (Appendix A). Microbial communities were dominated by Planctobacteria (63%), Verrucomicrobia (18%), Chloroflexi (7%), and Patescibacteria (6%) (Figure 1a).

When comparing plots treated with living vs. sterilized compost tea, there was no significant effect of fresh compost tea application on bacterial communities, neither through shifts in α-diversity (ASV richness: *p* = 0.64/Shannon’s diversity: *p* = 0.26/Inverse Simpson’s diversity: *p* = 0.56), nor shifts in community structure (β-diversity; perMANOVA pseudo-*F* = 1.17, *d.f.* = 2, *p* = 0.216; Figure 1b,c).

Indicator species analysis revealed that: (1) there were many indicator bacterial ASVs of compost tea (in fact, the majority of our indicator taxa belonged to compost tea); (2) only one ASV belonging Planctobacteria, was found to be an indicator species of both compost tea and treated soil samples; (3) several ASVs were indicators species of both treated and control plots; (4) control plots or only treated plots did not share any common indicator species (Figure 2).

### 2.2. Soybean Growth and Productivity 

Compost tea application did not improve plant growth (shoot dry mass, *p* = 0.36) or grain yield (grain dry weight, *p* = 0.14; Figure 3). Statistical power analyses indicated that compost tea application had small effect sizes (power = 23% and 30%, respectively, for growth and yield). We estimated that minimal sample size to detect an effect would have been 28 blocks for plant growth, and 20 blocks for plant yield, confirming the small effect size of our compost tea treatment.

## 3. Discussion

Surprisingly, both the control and treated soil samples were largely dominated by Planctobacteria (Figure 1a), a result contrasting with several studies identifying Proteobacteria as the dominant bacterial phylum in soils, followed by Chloroflexi, Bacteroidetes, Actinobacteria and Acidobacteria [43,44,45,46,47]. Planctobacteria are a unique divergent phylum of aquatic bacteria [48,49,50,51,52,53], that can be isolated from nonaquatic environments such as soil [52,54]. These bacteria are assumed to prefer anaerobic soil micropores [55,56,57], as they can tolerate low O_2_ pressures, which allows them to displace obligately aerobic taxa in low-O_2_ microsites/horizons [55,56]. Here, we hypothesize that soil dominance by Planctobacteria could be explained by the recent installation of drainage infrastructures in the subsoil horizon of our study site. This caused the mixing of topsoil with deep subsoil (2 m deep), which was presumably (1) less aerated and (2) less colonized by roots, which accordingly would account for the low abundance of copiotrophic rhizosphere specialists belonging to the Proteobacteria phylum [42]. This would be in line with Kepel et al. [58], who recently found that the only soil in their dataset dominated by Planctobacteria was from a rice field, which are typically characterized by low soil O_2_ pressures.

Because Planctobacteria were also abundant in the compost tea preparations (Figure 1a), we could also hypothesize that the Planctobacteria DNA retrieved in our soil samples (both from treated and control plots) belonged to dead bacterial cells, and this DNA had not fully degraded at the time of soil sampling. However, considering the total volume of liquid applied per surface in our treatments, we would find it surprising if the non-degraded portion of this dead DNA constituted the majority of the DNA we extracted afterwards from our soil samples. Moreover, by looking specifically at Planctobacteria communities in our soil and tea samples (i.e., by filtering our ASV table so that only Planctobacteria remain), we find that distinct Planctobacteria taxa dominated tea samples vs. treated soil samples (Appendix A).

Our principal component analysis (Figure 1c) revealed a clustering of bacterial communities according to their plot origin rather than their treatment (i.e., living vs. sterilized tea), which further shows that bacterial populations were spatially heterogeneous at our site, but not influenced by the treatment. This could be explained, in part, by contrasting soil properties across blocks (e.g., N availability; see Table 1).

The molecular analysis of bacterial community structure overall suggests a poor establishment of microbial taxa from the tea in the soil. This is supported by the fact that only one out of 737 ASVs was commonly found in both compost tea samples and treated soil samples (Figure 2). As our sensitivity analysis revealed that type I errors could represent around 3–4% of the dataset, we cannot rule out the possibility that the single indicator ASV for tea and for treated plots resulted from a type I error and thus was not truly an indicator for tea and treated plots. In fact, of our 737 ASVs, 322 were identified as indicator taxa (44%). This is well above the random expectation of 3–4%, but still, this means that roughly 10% of our indicator taxa may have arisen in the analysis by chance alone. However, this does not affect our conclusions, as most indicator taxa were indicators of either tea (probably taxa from tea that failed to thrive in the soil) or soil alone (resident soil taxa present prior to application). In both cases, this would suggest a poor establishment of tea bacterial taxa in our plots. Overall, this offers compelling evidence for the hypothesis that in our study, the tea bacteria failed to establish themselves in the soil, either because of low application density (and thus low initial population sizes) and/or because of a poor competitive ability against resident soil bacteria.

Compost tea application did not improve plant growth or yield in this experiment. Statistical power analyses confirmed the small effect size of our compost tea treatment, thus any impact of the compost tea on the living soil community (bacterial or not) would have been modest and would not have translated to large shifts in crop performance. Other studies performed on vermicompost tea or vermicompost leachate reported plant growth benefits and increased tolerance against abiotic stress. For example, Chaichi et al., 2018 [59] applied different concentrations of vermicompost tea ranging from 0% to 20%, as a foliar amendment in faba bean and they found that treated plants grow better than controls and had more flowers, clumps and pods per plant than non-treated pants [59]. Treatment at a concentration of 10% of vermicompost tea showed better results that 20%. The authors attributed these growth benefits the presence of nutrients, humic acid and putative hormones in vermicompost tea [59]. Other reports showed vermicompost leachate, a method consisting of adding water to a large amount of vermicompost (e.g., 20 L of water for 50 kg of vermicompost) allowing the mixture to leachate for 48 h, alleviated salt stress in tomato plants which was also explained by phytohormones and presence of other compounds such as humic acid and antioxidants [60,61]. Further experiments of comparison of compost tea, vermicompost tea and vermicompost leachate on plant growth and microbial communities of rhizosphare and phylosphese will bring insights on the effects of these natural amendments on crop growth and yield in field conditions.

## 4. Materials and Methods

### 4.1. Site Description

Our study was conducted in a field of approximately 3 hectares, located in Sainte-Christine, Quebec, Canada (see Appendix A 72.434353 W, 45.613667 N). This field has a several-decade history of conventional soybean-maize monocrop rotations and conventional agricultural practices. In spring 2018, installation of a drainage system in the field resulted in a severe soil disturbance in which the plow zone was mixed with the less biologically active, deeper horizons [12]. On 6 June, soybean was sown at a density of 382,850 seeds/ha.

### 4.2. Experimental Design

We conducted our compost tea application using a randomized block design. We divided the field (344 m by 82.5 m) into six experimental blocks (172 m by 27.5 m). Each block was then divided into two plots, with one receiving the treatment (living compost tea) and the other receiving the control (compost tea sterilized by boiling). Before application, we characterized initial soil properties by collecting composite soil samples from each block (Table 1).

### 4.3. Compost Tea Preparation and Application

Aerated compost tea was prepared in two phases: a pre-activation phase, aiming to increase microbial population densities in the compost, and a dilution phase, producing a liquid suspension from the compost (i.e., tea) for inoculation.

In the pre-activation phase, two different kinds of compost were mixed in equal quantities: the first, an especially carbon-rich vermicompost, consisting of up to 50% ramial wood chips and leaf litter and matured through the activity of earthworms, and the second, a thermal compost, consisting of 10% chicken manure, 15% horse manure, 30% fresh green plants, 20% ramial wood chips and 25% leaf litter, mixed at high temperatures (60–70 °C) for 30 days. In a 75 L container, we combined 20 L of this compost mixture with 300 mL oatmeal, 150 mL alfalfa flour, 150 mL fish hydrolysate, 100 mL seaweed flour, 30 mL molasses, 5 mL humic acid solution and non-chlorinated water (to reach 50% humidity). This blend was incubated for 72 h and mixed every 12 h to maintain aerobic conditions. Compost tea was prepared by washing this aerated mixture at room temperature in 20 L washing bags (mesh size = 400 µm), and then combining 10 L of the aerated mixture with 0.8 L water, 3 L oatmeal, 2.5 L fish hydrolysate, 1.5 L humic acid solution and 0.5 L soluble algae. Air was pumped into the mixture for two days to avoid anaerobic fermentation. Half of this compost tea preparation was then sterilized by heating at 95 °C for 90 min, in order to be used as an experimental control (i.e., to distinguish the effects of the compost tea’s living organisms from the abiotic effects of its minerals and dissolved organic nutrients).

The compost tea and sterilized control solution were prepared and applied to the field 4 times during the summer of 2018, on 9 June, 22 June, 5 July and 19 July. Dilutions and dosages were adapted to weather conditions during the growing season, thus the compost tea dilution ratios for the specified dates were 1:1, 1:4, 1:4 and 1:3, respectively, with dosage densities of 121.57 L/ha, 486.26 L/ha, 486.26 L/ha and 364.7 L/ha, respectively. In addition, subsamples of each of the compost tea preparations (i.e., concentrated, applied and sterilized) were kept at −20 °C for molecular characterization of the bacterial communities.

### 4.4. Field Samplings

Two field samplings were conducted. A first sampling campaign was performed during the vegetative growing stage, on 14 August, in order to (1) measure the aboveground dry biomass as an indicator of vegetative plant growth, and (2) characterize the bacterial communities present in the soils. In each plot, nine individual soybean plants were excavated and their rhizospheric soil collected (by shaking the root system in a plastic bag) and kept frozen at −20 °C for DNA extraction. Aboveground biomass was dried (at 65 °C for 1 wk) and weighed. The second field sampling was conducted a day before crop harvest, on 3 October, when 30 individual soybean pods per plot were randomly collected and transferred to the laboratory to measure grain weight as an indicator of yield.

### 4.5. Molecular Analyses

DNA was extracted from 250 mg of rhizospheric soil and compost tea samples using a Power Soil DNA kit (Qiagen Inc., Montreal, QC, Canada) according to the manufacturer’s instructions. Double-stranded DNA was quantified using a Qubit^®^ 2.0 Fluorometer (Thermo Fisher Scientific Inc., Ottawa, ON, Canada). DNA extracts were PCR-amplified using 16S rDNA primers with CS1 and CS2 adapters shown in italics (forward CS1-341F: 5′-*ACACTGACGACATGGTTCTACAC*CTACGGGNGGCWGCAG-3′; reverse CS2-806R: 5′-*TACGGTAGCAGAGACTTGGTCTGACT*ACHVGGGTATCTAATCC-3′ [62]), targeting the hypervariable V3-V4 region of the 16 rRNA gene. PCR reactions were performed in a total volume of 50 µl containing 1X PCR buffer, 0.5 µM of each primer, 5.0 µL of dNTPs (10 mM), 0.4 µL of Taq DNA polymerase and 2 µL of template DNA. PCR conditions were as follows: 4 min denaturation at 94 °C, followed by 35 cycles of denaturation (94 °C for 30 s), annealing (55 °C for 30 s) and extension (72 °C for 60 s), and a final 10 min extension at 72 °C. PCR reactions that gave a visible amplification band on agarose gel were sent for Illumina MiSeq sequencing (300 bp paired-end library) at the Génome Québec Innovation Center.

### 4.6. Bioinformatics

Analysis of the sequence data was coded in R (v4.0.1; R Development Core Team, 2014) using the *DADA2* R package (v1.1.2; [63]). Sequences were quality filtered and primers were removed. We removed sequences with less than 290 bp and 260 bp (forward and reverse, respectively), as the base quality of the sequences showed a clear drop below these thresholds. For this we used the DADA2 command *filterAndTrim* with a maxEE score of 2 and trunQ score of 6. We then calculated the error rates using the *learnErrors* command and merged the forward and reverse sequences. Next, chimeras were removed and the amplicon sequence variants (ASV) table was built, and finally taxonomy was assigned to the ASVs using the SILVA reference database [64].

A total of 2,171,433 raw reads were generated from 119 individual samples (108 soil samples and 11 compost tea samples). Sequences classified as chloroplasts or mitochondria were removed from the ASV table, as were any sequences classified as Eukarya or Archaea. Samples were then rarified to 1749 reads per sample using the function *rrarefy* from the R package *vegan* [65]. To avoid focusing on potential sequencing artefacts or on especially rare bacterial taxa, we filtered our dataset to remove: (1) any occurrences with 5 reads, and (2) any ASVs that only appeared in 1 or 2 samples.

### 4.7. Statistical Analysis

To determine the effect of compost tea and sterilized compost tea (control) treatments on plant growth and yield, we used linear mixed models (LMMs) as implemented in the R package *lme4*, including plot identity as a random effect [66]. We used the *pwr* R package to estimate the power of our analysis comparing the plant growth or yield of treated plots vs. control plots [67].

We evaluated the impact of compost tea application on both bacterial α-diversity and community structure (β-diversity). Alpha diversity was assessed using ASV richness, the exponential form of Shannon diversity and inverse Simpson diversity [68]. Alpha diversity was compared between treatments using Poisson regression (generalized LMM) for ASV richness and Gaussian LMMs for Shannon and Simpson diversities.

Shifts in community structure following treatments were assessed with permutational multivariate analysis of variance (perMANOVA; [69]) using the function *adonis* of the R package *vegan* [65]. The Hellinger distance [70] was used to evaluate pairwise β-diversity across samples, which was visualized using principal component analysis (PCA).

In order to identify bacterial ASVs that were specifically associated either with soil samples treated with compost tea or control samples treated with sterilized compost tea, we conducted an indicator species analysis (ISA) using the function *multipatt* of the R package *indicspecies* [71]. We used a threshold of α = 0.01 because this analysis implied a high number of taxa in permutation-based statistical tests (i.e., 1 per bacterial taxon), which may inflate type I errors. However, traditional *p*-value correction methods (e.g., Bonferroni) would have resulted in overly conservative tests given the very high number of bacterial ASVs. This would have resulted in high type II error rates, which is why we decided to manually set α at 0.01. To evaluate how prone we were to detecting false positives (i.e., indicator taxa that would be associated with one treatment or another simply by chance), we conducted ISA on randomized metacommunities generated using the null model *vaznull* in the R package *bipartite* [72]. These simulations indicated that roughly 3–4% of indicator taxa may arise as false positives (Appendix A).

## 5. Conclusions

Our results showed that aerated compost tea application had no influence on bacterial diversity or community structure. Accordingly, plant growth and yield were unresponsive to compost tea application. We note that our results do not undermine the potential role of compost tea in increasing crop yield or contributing to sustainable agriculture. Our design did not include plots where compost tea was not applied. Our study thus reveals that the positive effects of compost tea found in other studies could be due to: (1) the nutritional effects of compost tea (i.e., its contribution of minerals to the soil through dissolved organic nutrients), or (2) its alteration of other physicochemical properties of the soil (e.g., increased cation exchange capacity due to dissolved organic matter in compost). Alternatively, the absence of effects in our study could be ascribed specifically to the compost we used or to the dose or frequency of tea application. Much remains to be studied regarding the mechanistic nature of the impact of compost tea on crop performance. Our only conclusion here is that in this field trial on a severely disturbed soybean monoculture field, living compost tea application did not influence bacterial communities or crop yield.

## Figures and Tables

**Figure 1 plants-10-01638-f001:**
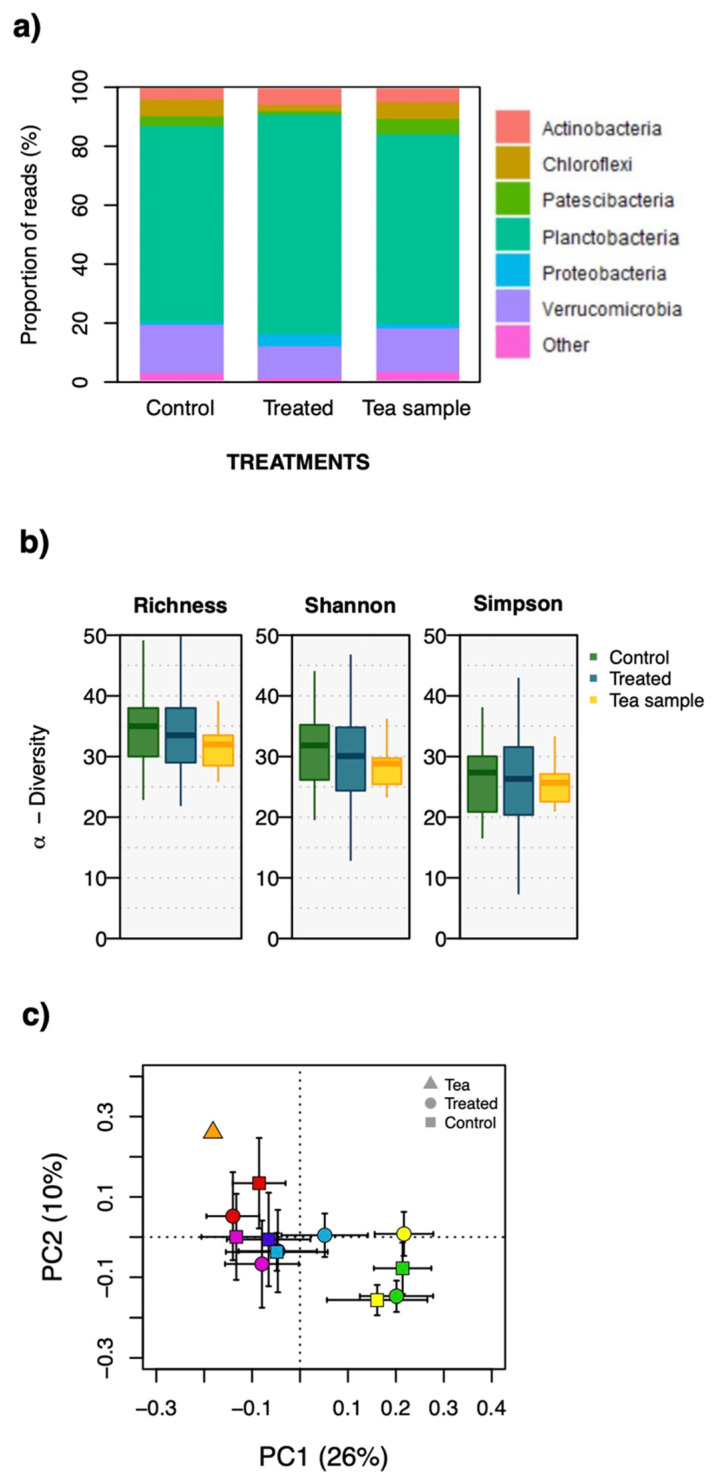
(**a**) relative abundance of ASVs belong to the different bacterial phyla present in soils treated with living compost tea (“Treated”), with sterilized compost tea (“Control”) or from tea samples taken prior to application (“Tea sample”). (**b**) alpha-diversity of bacterial communities (exponential Shannon (*eH*) and inverse Simpson indices). (**c**) principal component analysis (PCA) of Hellinger-transformed bacterial relative abundances. Bacterial communities tend to cluster according to experimental blocks (yellow, green, dark-blue, light-blue, pink and red, represent six experimental blocks; orange triangle represents compost tea). Shapes (circle and square) represent treatments. Symbols represent the mean scores of samples from a given plot, and error bars represent 95% confidence intervals.

**Figure 2 plants-10-01638-f002:**
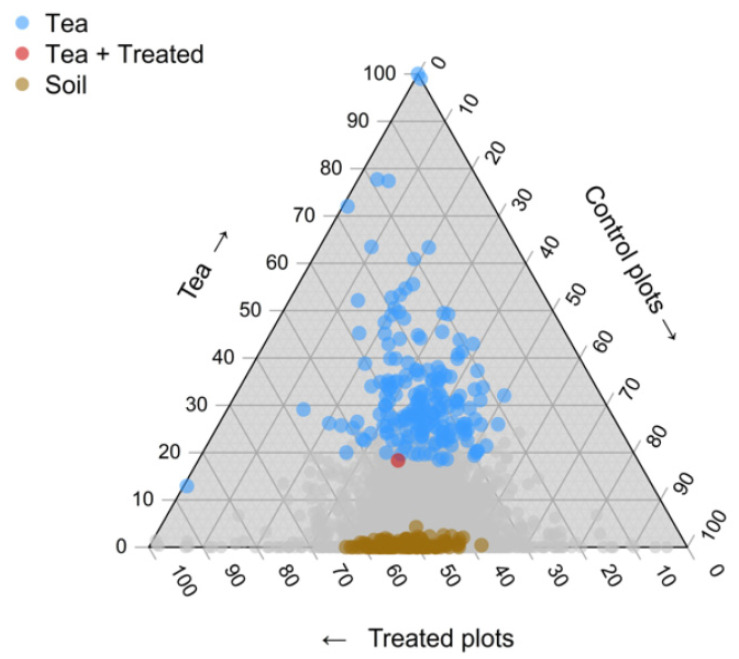
Ternary triangle presenting the relative distribution of reads from each ASV in treated plots, control plots or tea samples. Each symbol represents an ASV. Blue symbols are ASV indicator for tea samples; brown symbols are ASV indicator for control plots; the red symbol is the only ASV indicator for both tea and treated plots; grey symbols are those ASVs that are not indicator for any category.

**Figure 3 plants-10-01638-f003:**
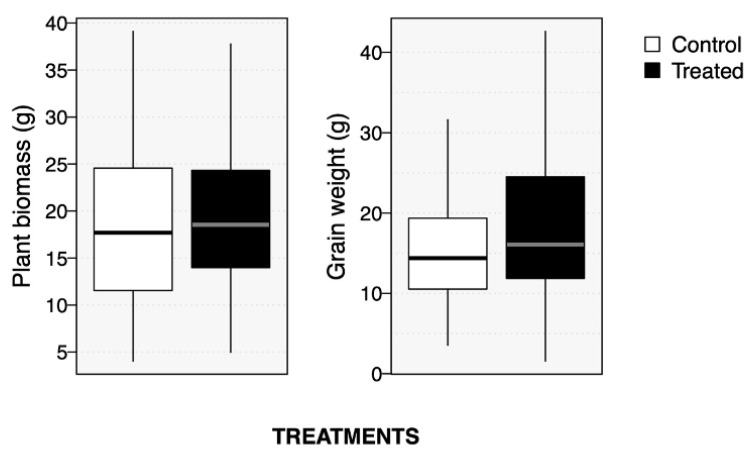
Boxplots showing plant growth (left) or yield (right) in control plots (red) or treated plots (cyan).

**Table 1 plants-10-01638-t001:** Soil properties were measured on composite samples taken from each experimental block. Mehlich III-PO_4_^3−^ = orthophosphates extractible with Mehlich-III solution; KCl-NH_4_^+^ and KCl-NO_3_^−^ are, respectively, ammonium and nitrates extractible using 2N KCl.

Block	pH	Organic Matter Content (%)	Gravimetric Moisture (%)	Melich III-PO_4_^3−^ (mg/kg)	KCl-NH_4_^+^ (mg/kg)	KCl-NO_3_^−^ (mg/kg)
A	6.20	12.75	25.26	75.61	68.91	5.84
B	6.25	9.30	20.79	26.28	62.74	8.44
C	6.16	14.02	22.53	26.60	81.77	9.38
D	6.36	10.27	24.46	32.73	91.31	20.43
E	6.64	6.88	21.42	32.06	28.38	14.04
F	6.56	12.21	25.16	24.26	44.48	19.89
Mean	6.36	10.91	23.27	36.26	62.93	13.01

## Data Availability

The datasets generated and analyzed during the current study are available in the Sequence Read Archive (SRA) under project number PRJNA728448. [http://www.ncbi.nlm.nih.gov/bioproject/728448].

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
