# Peer review of "Fresh Compost Tea Application Does Not Change Rhizosphere Soil Bacterial Community Structure, and Has No Effects on Soybean Growth or Yield"

_plants, 2021, doi:10.3390/plants10081638_

Round 1
Reviewer 1 Report
The paper presents the results of an independent study on the possible application of tea fresh compost on soil bacterial community structure. Nowadays preservation of soil structure and microorganism composition is one of the main challenges for growers, due to an intensive conventional agriculture that lead to deletion of nutritive elements and microbial diversity of the soil. To elucidate the possible use of compost tea as bioinoculants, the authors conducted a field experiment in a conventional soybean monoculture system, anayzing the bacterial community in the inoculated soil and the growth of treated plants. It is interesting the presentation and discussion of results that are not in agreement with previous research papers, sshowing that working with living organisms in different natural conditions could influence the obtained data. Based on all these consideration I wold suggest to publish he paper after minor spelling check, in order to open new perspectives on the topic.
Author Response
We were very pleased to hear that the reviewer#1 found the manuscript to be interesting and appropriate for Plants. We are very grateful this reviewer for taking the time to think about this work seriously.
In view of the comments made bu Reviewer#2, we have modified the text of some sections, added new supplementary Figure S1 and corrected figure labels and improved figure quality. We added information on rarefaction analyses and provided Fig. S1 to answer those criticisms made by Reviewer#2.
Reviewer 2 Report
This study investigated the effect of the application of tea compost on the soil bacterial diversity in a conventional soybean monocolture system . The experiment was carried out in a field with a soybean-maize monocrop rotation and conventional agricultural practices history. The soil bacterial diversity and structure was analysed by Illumina MiSeq sequencing, whereas soybean growth and productivity were analysed by the measures of the shoots dry mass and grains dry weight, respectively. Results revealed that the application of tea compost do not improve soybean growth and productivity and do not influence the biodiversity and structure of the bacterial communities associated to the roots of soybean plants.
In general, the manuscript reads well and the experimental field design is good. However, there are several points that need to be explained.
My main concerns are on the procedure used to prepare the tea compost and on the number of reads obtained by MiSeq sequencing.
Was the tea compost prepared accordingly to the procedure applied in other works in which it was used to improve plant growth and productivity? Moreover, the chemical characterization of tea compost is missing. This is relevant because it is important to compare the results with those obtained in other studies.
A total of 119 samples were subjected to MiSeq-Illumina sequencing. From these samples, 2,171,433 raw reads were generated and 1749 reads for each sample were generated after excluding low-quality sequence reads, preclustering, and the chimeras removal. Are you sure that these reads explain all the bacterial diversity present in each sample?
Are there no differences in the number of reads obtained among the treatments? Could you please provide the rarefaction curves obtained for the different treatments? A figure showing the rarefaction curves could be included in the supplementary material.
I also suggest to change in the title the word “soil” with “rhizosphere soil”. Indeed, you sampled the soil surrounding the roots of the plants.
Main specific comments:
Line 34. Which type of biodiversity, other than bacterial diversity, do you refer to?
Line 82. Please add “microbial” before “communities”.
Figure 1. Please change “Proportion of reads” both in the figure 1.a and the legend with “Relative abundance of ASVs”. In fig. 1.a, it is necessary to change the color of the Planctomyces because it is similar to that of the Proteobacteria phylum.
Line 88. Please eliminate “samples, controlled soil samples (Control) and tea samples”.
Line 89.Eliminate “respectively”. What do you mean for exponential Shannon and
inverse Simpson diversity? Could you simply write exponential Shannon and inverse Simpson indices?
Line 91. It is not clear the relation of symbol colors with the experimental block. Please specify.
Lines 99-104. This part is not clear.
Line 133. Change “in” with “belonging to the Proteobacteria phylum”.
Line 147. Please change “our” with “The”.
Lines 153-155. This part is not clear.
Line 157. Please change “our” with “the”.
Table 1. “Soil properties were measured on composite samples taken from each experimental block. Mehlich III – PO43- = orthophosphates extractible with Mehlich-III solution; KCl-NH4+ and KCl-NO3- = respectively ammonium and nitrates extractible using 2N KCl.” This should be the legend of the table.
Author Response
We were very pleased to hear that the reviewer#2 found the manuscript to be interesting and appropriate for Plants. We found her/his comments very helpful to make a major revision of the manuscript that we feel answered all the criticisms and makes it a much more solid and rigorous manuscript. We are very grateful to this reviewer for taking the time to think about this work seriously and provide us with the opportunity to present a more concise paper. In view of the comments, we have modified the text of some sections, added new supplementary Figure S1 and corrected figure labells and improved figure quality. We added information on rarefaction analyses and provided Fig. S1 to answer those criticisms. This does not change the main conclusions of our study but soften some of them, which make the paper much stronger.
I have given the specific details (red) in the Response to reviewer#2 of how the manuscript has been changed or how we have addressed the reviewer’s comments.
In general, the manuscript reads well and the experimental field design is good. However, there are several points that need to be explained.
We thank Review#2 for providing us with helpful comments and suggestions.
My main concerns are on the procedure used to prepare the tea compost and on the number of reads obtained by MiSeq sequencing.
Was the tea compost prepared accordingly to the procedure applied in other works in which it was used to improve plant growth and productivity? Moreover, the chemical characterization of tea compost is missing. This is relevant because it is important to compare the results with those obtained in other studies.
Our compost tea was prepared according to the procedure applied in other works with some modifications of ingredients. The recipes of compost tea preparation may differ between studies, application purposes and crops. The protocol used in our study was developed in-house using diversified ingredients. Because all ingredients used to prepare our compost tea were known, we didn’t perform chemical analyses of diluted compost tea before applications.
A total of 119 samples were subjected to MiSeq-Illumina sequencing. From these samples, 2,171,433 raw reads were generated and 1749 reads for each sample were generated after excluding low-quality sequence reads, preclustering, and the chimeras removal. Are you sure that these reads explain all the bacterial diversity present in each sample?
Are there no differences in the number of reads obtained among the treatments? Could you please provide the rarefaction curves obtained for the different treatments? A figure showing the rarefaction curves could be included in the supplementary material.
We thank the reviewer for this comment and suggestion. As you can see, we added information of rarefaction analysis of all samples. A new Figure S1 showing rarefaction analysis was added in supplementation figures (we changed accordingly the order of Suppl. Figures). Rarefaction analysis showed that all curve reached saturation and the number of ASVs was adequate.
I also suggest to change in the title the word “soil” with “rhizosphere soil”. Indeed, you sampled the soil surrounding the roots of the plants.
We added rhizosphere soil in the title.
Main specific comments:
Line 34. Which type of biodiversity, other than bacterial diversity, do you refer to?
We rephased the sentence to include community structure and biomass.
Line 82. Please add “microbial” before “communities”.
Added
Figure 1. Please change “Proportion of reads” both in the figure 1.a and the legend with “Relative abundance of ASVs”. In fig. 1.a, it is necessary to change the color of the Planctomyces because it is similar to that of the Proteobacteria phylum.
We replaced “Proportion of reads” both in the figure 1.a and the legend with “Relative abundance of ASVs”. We also changed color codes of Fig. 1a.
Line 88. Please eliminate “samples, controlled soil samples (Control) and tea samples”.
Removed
Line 89.Eliminate “respectively”. What do you mean for exponential Shannon and
We removed “respectively”. We mean exponential of Shannon entropy index (eH)
inverse Simpson diversity? Could you simply write exponential Shannon and inverse Simpson indices?
Changed with (exponential Shannon (eH) and inverse Simpson indices).
Line 91. It is not clear the relation of symbol colors with the experimental block. Please specify.
We rephrased legend for clarity and add information for colors and shapes used in Fig. 1C as well as in Fig. S3.
Lines 99-104. This part is not clear.
We rephrased the sentence for clarity.
Line 133. Change “in” with “belonging to the Proteobacteria phylum”.
Changed
Line 147. Please change “our” with “The”.
Done
Lines 153-155. This part is not clear.
The sentence was rephrased as: “This is supported by the fact that only one out of 737 ASVs was commonly found in both compost tea samples and treated soil samples (Fig. 2)”
Line 157. Please change “our” with “the”.
Changed
Table 1. “Soil properties were measured on composite samples taken from each experimental block. Mehlich III – PO43- = orthophosphates extractible with Mehlich-III solution; KCl-NH4+ and KCl-NO3- = respectively ammonium and nitrates extractible using 2N KCl.” This should be the legend of the table.
Changed